# Influence of W Addition on Microstructure and Mechanical Properties of Al-12%Si Alloys

**DOI:** 10.3390/ma12060981

**Published:** 2019-03-25

**Authors:** Anna Zykova, Nikita Martyushev, Vadim Skeeba, Denis Zadkov, Andrey Kuzkin

**Affiliations:** 1Tomsk Polytechnic University, 30, Lenina Ave., Tomsk 634050, Russia; martjushev@tpu.ru; 2Novosibirsk State Technical University, 20 Prospekt K. Marksa, Novosibirsk 630073, Russia; skeeba_vadim@mail.ru; 3Saint-Petersburg Mining University, 2, 21 Line of Vasilyevsky Island, Saint Petersburg 199106, Russia; dzadkov@yandex.ru (D.Z.); kuskinay@bk.ru (A.K.)

**Keywords:** Al-12%Si alloy, W, nanopowder, modification, microstructure, mechanical properties

## Abstract

A widespread method exerting the influence on the homogeneous formation of the microstructure and enhancement of strength properties of Al-Si alloys is a modification by super- and nanodispersed particles of different chemical compositions. In spite of the significant advances in the studies of the influence of various modifying compositions on the structure and mechanical properties of casted silumins, the literature contains no data about the influence of nanodispersed W-powder on formation of the structural-phase state and mechanical properties of Al-Si alloys. The paper considers the influence of 0.01–0.5 mass % W nanopowder on the structural-phase state and mechanical properties of an Al-12%Si alloy. It has been established that 0.1 mass % of W is an optimal addition. It results in the uniform distribution of eutectic (α-Al + Si), a 1.5-time decrease in the size of the plates of eutectic Si, a change of the shape of coarse plates (coarse plate-like or acicular) into a fine fibrous one, and an enhancement of the mechanical properties by 16–20%.

## 1. Introduction

Casting alloys of the Al-Si system are extensively used as modern structural materials in aircraft and automobile industries and others owing to their high specific strength, resistance to dynamic loads and low cost. Nevertheless, their extensive use in industry is often limited because of the presence of coarse-crystalline structures consisting of the α-Al solid solution, eutectic (α-Al + Si) and Fe-containing phases in alloys. The quality of silumins and their mechanical properties depends not only on the chemical composition, but in many respects they are determined by uniformity and dispersity of the cast products. The structure of the alloy under formation determines the strength properties and plasticity of silumins. Simultaneously, morphology of eutectic silicon and Fe-containing phases, formed during eutectic and secondary crystallization of melts, impact mechanical properties of the alloy. 

A popular method that affects the microstructure formation and enhances mechanical properties of Al-Si alloys is an introduction of additions of super- and nanodispersed particles of various chemical compositions into the melt. Nowadays, a lot of experimental data on the use of metal oxides (γ-Al_2_O_3_, Al_2_O_3_, TiO_2_, ZrO_2_, SrO) [1,2,3,4,5], rare-earth elements (Sr, Cr, Sm, Eu, etc.) [6,7,8,9], carbides (TiCN) [10], borides (Nb-B, Ti-B) [11,12] have been accumulated for changing dendritic α-Al, the morphology and the size of eutectic Si and Fe-containing phases.

The urgency of the present paper is conditioned by the fact that, despite the significant advances in the research on impact of different modifying compositions on the structure and mechanical properties of casted silumins, there are no data about the influence of the nanodispersed W powder on the formation of structural-phase state and mechanical properties of Al-Si alloys. In connection with this, in general, ligatures and sintered briquettes of powders of different chemical elements (Ti, B, W, Ta, Zr, V, Mo, Cr, P, Be, S, Ce, etc.) are used for the modification of silumins [13], although there are a number of papers on silumins modification by means of introduction of nanoparticles into the melt [10]. Therefore, the aim of the present paper was to conduct research into the influence of W nanopowder additions on the microstructure and mechanical properties of Al-12%Si alloys.

## 2. Materials and Methods 

The nominal composition of the Al-12%Si alloy employed in the present study is listed in Table 1.

The W nanopowder obtained by means of electrical explosion of conductors with the average size of the particles being *d* = 130 nm and specific surface area—S_BET_ = 1.7–2.4 m^2^/g was used as a modifying agent. According to XRD and SEM data, the powder particles had a spherical shape and consisted of 95 mass % of W and 5 mass % of W_3_O (Figure 1).

The Al-12%Si alloy was melted in a muffle furnace. To force the melting process, the furnace was at first heated up to 800 °C, and then a steel crucible with the mix material was placed into the furnace. After melting the Al-12%Si alloy, the crucible was withdrawn from the furnace, the surface of the melt was cleaned from the oxide film and then W was introduced. The amount of the W nanopowder introduced into the melt was 0.01, 0.05, 0.1, 0.5 mass %. The W powder was wrapped in aluminum foil and introduced into the melt. Then, the melt was stirred during 5 min by a ceramic stirrer. The melt of the Al-12%Si alloy with the modifying W addition was held in the furnace at a temperature of 800 °C during 10 min. The pouring was conducted at a rate of 0.06–0.09 l/s. The casting of samples was carried out into a cast-iron mold with the 1:10 ratio of the casting mass to the mold mass. The cooling rate at the moment of crystallization was approximately 80–90 °C/s.

The phase compositions of nanodispersed W powders and the Al-12%Si alloy were determined by X-ray diffraction (XRD) using an X-ray diffractometer ‘Rigaku Miniflex 600’ (Thermo Fisher Scientific, Waltham, MA, USA) with CuKα-radiation at the scan rate of 2 °/min. Morphological characteristics of the tungsten nanopowder were studied by scanning electron microscopy (SEM) using a microscope ‘JSM 7500F’ (JEOL, Tokyo, Japan). The localization phase of the Al–12%Si alloy and the basic structural characteristics were studied using an optical microscope (ALTAMI-MED 1C, Altami, Saint Petersburg, Russia) and a scanning electron microscope ‘Tescan Vega II LMU’ (TESCAN, Kohoutovice, Czech Republic), compatible with energy dispersive X-ray spectroscopy (EDX) ‘INCAx-act’.

The crystallization processes were studied by differential scanning calorimetry (DSC) using a thermoanalyzer ‘SDT Q600’ (TA Instruments, New Castle, DE, USA) at a heating/cooling rate of 10 °C/min. 

The hardness of as-cast samples was determined according to Brinell using a Brinell hardness tester ‘TSH-2M‘ (Microtester, Moscow, Russia) with a load of 250 kg and a steel ball with a diameter of 5 mm. The measurements were conducted on the undeformed surfaces of shock samples controlling the parallelism of two sides. The surface under test was preliminary ground and polished.

The ultimate tensile strength and the relative elongation were measured using a modern testing machine ‘MIRI-100K’ (Tochmashpribor, Armavir, Russia) with computer control and calculation of mechanical characteristics. Strength characteristics during tension were determined using the scheme of uniaxial tension on cylinder samples with fillets of type III with a diameter of 6 mm in the working section; the diameter of the fillets (gripping parts) was 10 mm. The strain rate was 0.2 mm/s.

For carrying out impact toughness tests, the impact pendulum-type testing machines of models ‘MK-30А’ and ‘2130KM-0.3’ (Experimental plant "IMPULSE", Ivanovo, Russia) with a maximum energy of the pendulum equal to 300 J were used. Standard Charpy unnotched specimens with dimensions of 55 mm × 10 mm × 10 mm were used for the tests. The number of specimens was 3–6 for each series of tests.

## 3. Results

### 3.1. Effect of W Addition on Phase State and Microstructure of Al-12%Si Alloy

The quality of castings of Al-12%Si alloys before and after the introduction of W additions was estimated according to the α-Al dendrite structure, the morphology and the size of silicon plates (Figure 2). The microstructure of the initial Al-12%Si alloy is characterized by α-Al dendrites with axes of the third order and a coarse (α-Al + Si) eutectic with a coarse needle-like structure of Si (Figure 2a). According to the data of optical microscopy, the distance between the α-Al axes of the second order is 28–32 μm; at that, the length of axes of the first order of separate crystals reaches 2.5–3 mm. The morphology of eutectic Si is characterized by the coarse needle-like structure, the average size is 12.3 µm (Figure 2a). In the castings of the initial Al-12%Si alloy, the clusters of eutectic (α-Al + Si) are observed.

An introduction of the nanodispersed tungsten powder in the amount of 0.01–0.5 mass % into the melt of the Al-12%Si alloy influenced the castings microstructure. When introducing 0.01 and 0.05 mass % of W into the melt, the microstructure of the resulting alloy is characterized by the α-Al dendrites with axes of the second order similarly to the initial alloy. At the same time, the axes of the third order are practically absent, and the distance between the axes of the second order of α-Al dendrites is reduced to 18–22 μm (Figure 2b,c). In the castings, the clusters of eutectic (α-Al + Si) and the presence of coarse-dispersed Si wafers are also observed; along the boundaries, fine Si particles with the size of about 1 μm are found. The average size of Si wafers in eutectic when introducing 0.01 mass % of W is 9.6 µm, and when introducing 0.05 mass % of W—6.5 µm, which is substantially less compared to the initial castings (Figure 2a–c).

The introduction of 0.1 and 0.5 mass % of W into the melt leads to a uniform distribution of α-Al dendrites of the first order and (α-Al + Si) eutectic over the section of the casting (Figure 2c,d). If in case of the initial alloy, the distance between the axes of the second order is 25–35 µm, this value reduces to 15–25 µm, when introducing 0.1 and 0.5 mass % of W. The morphology of eutectic silicon also changes: the shape of coarse wafers changes into a fine fibrous one. The average size of Si wafers in eutectic is ~8 µm (Figure 2d,e). A similar effect of changing the shape of coarse-dispersed wafers of eutectic Si into a fine fibrous one is achieved when introducing rare-earth elements: 0.5 wt % of Eu [8], 0.6–1 mass % of Sm [9,14], 0.014–0.5 wt % of Sr [6,15], Er [16], La [17,18], Ce [8]. It is worth noting that the content of different additions of rare-earth elements to Al-Si alloys producing a modifying effect does not exceed 1%.

According to the optical microscopy data, the iron-manganese phases in a Chinese script-like form and in the form of plates have been discovered in the structure of Al-12%Si alloys along the boundaries of α-Al grains and in the (α-Al + Si) eutectic (Figure 3). The size of the α-(AlSiFeMn) phase is ~1.5–28µm (length), and that of the β-(AlSiFeMn) phase—4–45 µm (length). After the introduction of 0.01 mass % of W, the strongest change of the morphology of α-(AlSiFeMn) and β-(AlSiFeMn) phases occurs. Iron-manganese α-(AlSiFeMn) and β-(AlSiFeMn) phases acquire a compact with a size of ~2.8 μm (length) (Figure 3b). It is impossible to determine their modification based on metallographic data: whether it is α- or β-. However, when introducing 0.1 mass % of W into Al-12%Si alloys, the modification of iron-manganese α-(AlSiFeMn) and β-(AlSiFeMn) phases retains, but the size is reduced by more than 2 times (Figure 3c). More detailed studies on the influence of the W nanopowder on the morphology and the stoichiometric composition of iron-manganese α-(AlSiFeMn) and β-(AlSiFeMn) phases in Al-12%Si alloys will be presented in subsequent manuscripts.

According to the XRD, Al-12%Si alloys before and after the introduction of the W nanopowder are characterized by a solid α-Al solution and a Si phase. Other phases were not detected by this method (Figure 4). After the introduction of the tungsten nanopowder, only changes in the preferred orientation of the grains can be observed in the X-ray diffraction patterns. The volume fraction of iron-manganese α-(AlSiFeMn) and β-(AlSiFeMn) phases in Al-12%Si alloys is small and is not identified against the background of the basic peaks.

Thus, the introduction of the W addition (0.01–0.5 mass %) influences the formation of eutectic Si compared to the initial alloy: reduces the wafers size by 1.5–2 times, and the shape of the coarse-dispersed wafers changes into a fine fibrous one. The introduction of 0.01–0.05 mass % of W results merely in the reduction of eutectic silicon plates; at the same time, as well as in the initial alloy, the clusters of (α-Al + Si) eutectic and their nonuniform distribution over the casting section are observed.

### 3.2. Effect of W Addition on Mechanical Properties of Al-12%Si Alloy

Changes in the microstructure of the Al-12%Si alloy after the introduction of 0.01–0.5 mass % of W have influenced the ultimate tensile strength, the relative elongation and the impact toughness of the alloys. Data on the influence of additions of the W nanopowder on the mechanical properties of Al-12%Si alloys are presented in Figure 5. The introduction of the W nanopowder in the amount of 0.01–0.05 mass % into the melt leads to an equivocal influence on the mechanical properties. In the Al-12%Si alloy containing 0.01 mass % of W, the relative elongation reduces by 0.6 % and impact toughness increases by 16%; at that, the ultimate tensile strength remains virtually unchanged. When introducing 0.05 mass % of W, the tensile strength increases by ~20%; at that, the relative elongation and the impact toughness remain unchanged.

Figure 5 shows that the introduction of the addition in the amount of 0.1 mass % is optimal relatively mechanical properties as it leads to the increase of the ultimate tensile strength, the relative elongation and the impact toughness by 16–20%. The increase of the tungsten powder content up to 0.5 mass % results in the increase of the ultimate tensile strength by 10%, but the relative elongation and the impact toughness remain without changes. The studies have also indicated that the introduction of the tungsten nanopowder into the melt does not influence the hardness of the Al-12%Si alloy. Independently of the amount of the introduced modifier, the hardness equals HB50-55.

The enhancement of mechanical properties of the Al-12%Si alloy when introducing W is related to a reduction of eutectic Si sizes. Figure 6a shows that with the increase of the W content, the average size of eutectic Si reduces. The enhancement of mechanical properties is also related to the morphology change and the more than 2-time reduction in sizes of Fe-containing phases as shown in Figure 6b. On the whole, the W introduction in the amount of 0.05–0.5 mass % did not influence the volume fraction of Fe-containing phases; however, when introducing 0.01 mass % of W, the volume fraction of Fe-containing phases reduces more than 2 times (Figure 6b). Figure 3b indicates that the morphology of Fe-containing phases changes significantly, and the size reduces 2 times, which correspondingly influences the volume fraction.

Huang X. and Yan H. [17] observed a similar effect in mechanical properties enhancement when adding 0.3–1 wt % of La in ADS Al-Alloy. Compared with unmodified ADC12, the tensile strength and elongation of the alloy with the addition of 0.3–0.6 wt % of La increased by 10.8% and 58.1%, respectively. This happens because of the change in Si appearance and grain refinement in the alloy. The introduction of more than 0.6 wt % of La into ADS Al-Alloy leads to deterioration of mechanical properties because of the loss of the RE-containing intermetallic compound, which showed that a coarse and acicular plate morphology was formed in the alloy. Li Q. et al. [8] also indicated that the ultimate tensile strength and elongation increased by 68.2% and 53.1%, respectively, when introducing rare earth cerium (up to 1 wt % of Ce) into the Al–20%Si alloy, due to decreasing the size and changing the morphology on primary and eutectic Si crystals.

The introduction of a large amount (more than 0.5 mass %) of the W powder, reduces the modification effect. The plausible reason of it is agglomeration of the powder particles among themselves. The final number of the crystallization centers as a result of agglomeration becomes less than that when a small amount of the powder (0.01–0.1 mass %) is introduced. The optimal concentration for the obtainment of enhanced mechanical characteristics is 0.1 mass %. With this amount of the tungsten nanopowder, the refinement of the structure occurs (the decrease of α-Al dendrites sizes of the solid solution), the sizes of needle-like silicon additions decrease, which positively affects the properties. It is worth noting that the traces of W are not detected by XRD, SEM and EDX methods. Additional research by transmission electron microscopy (TEM), using mapping and a less diameter of the beam of the electron gun of EDX analysis, is required for detecting the traces of tungsten particles.

### 3.3. Crystallization Processes in Al-12%Si Alloys with Different Contents of W

Conducting the DSC-analysis is an indirect confirmation of changing crystallization processes of initial and modified alloys. This analysis is applied for studying phase transformations and mechanisms in the crystallization process of Al-12%Si alloys by different modifying additions [9,19]. Thus, in order to analyze the effect of W additions on eutectic temperature of the alloys in the present work, the DSC-analysis was carried out. In Figure 7, there is one endothermic peak corresponding to eutectic Si observed on each curve.

As a result, it was established that in the initial Al-12%Si alloy, the crystallization process of (α-Al + Si) eutectic is triggered at a temperature of 590 °C (Figure 7). The crystallization process of Al-12%Si alloys after the introduction of the W nanopowder is characterized by the shift of the instant of the (α-Al + Si) eutectic precipitation onset to the region of high temperatures by ~9 °C on average, which is 598–600 °C compared to the initial alloy (Figure 7). It is worth noting that the temperature shift of eutectic precipitation by ~9 °C is typical for each introduced addition of W (0.01–0.5 mass %). Despite the fact that the liquid us temperature has changed merely by 9 °C, and the interval of crystallization has increased by 5 °C, there are significant changes during crystallization of eutectic Si (1.5 times reduction in size) and iron-manganese phases (2–4 times reduction in size). In addition, the peaks corresponding to the minor (α-(AlSiFeMn) and β-(AlSiFeMn)) phases are not observed in the DSC-curves, which may be attributed to their low content. In the paper [9], the authors also demonstrated that with the increase of the Sm content in the ADC12 alloy, the temperature of primary crystallization changes by 2 °C per each introduction of the addition, and the wafers of eutectic Si are refined without alteration of its morphology.

Thus, the studies have demonstrated that the W nanopowder is efficient for refinement of structural constituents of Al-Si alloys, as, being a refractory element, it forms points of the crystallization onset in the melt shifting the temperature of the eutectic precipitation onset to the region of higher temperatures and accelerating the cooling rate of castings and refining the structural constituents of the alloy.

## 4. Conclusions

In the paper, the influence of the W nanopowder in the amount of 0.01–0.5 mass % on the structural-phase state and mechanical properties of the Al-12%Si alloy has been studied. It has been demonstrated that the introduction of 0.01 and 0.05 mass % of W into the melt leads merely to the downsizing of eutectic Si plates and influences ambiguously the mechanical properties of the Al-12%Si alloy. Despite the reduction of Si sizes, the clusters of (α-Al + Si) eutectic in samples, as well as in the initial alloy, are observed. It has been established that 0.1 mass % of W is an optimal addition as it results in the uniform distribution of (α-Al + Si) eutectic, the 1.5 times reduction of the eutectic Si plates and the change of the coarse needle-like plates into a fine fibrous one. With such kinds of change in the castings structure, the ultimate tensile strength, the relative elongation and the impact toughness increase by 16–20%. The increase of the tungsten nanopowder content up to 0.5 mass % leads to a 1.5-time reduction of Si wafers sizes; at the same time, the tensile strength increases by 10%, but the relative elongation and the impact toughness remain unchanged. The refinement of structural constituents and enhancement of mechanical properties of the Al-12%Si alloy have been found to be related to the change of crystallization processes when introducing 0.01–0.5 mass % of W. The introduction of W additions into the melt leads to a shift of the instant of the eutectic (α-Al + Si) precipitation onset to the region of higher temperatures by ~9 °C on average, which allows accelerating the cooling rate of castings.

## Figures and Tables

**Figure 1 materials-12-00981-f001:**
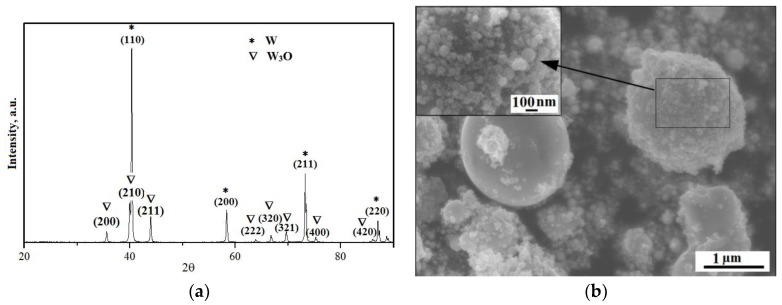
W nanopowder: X-ray difraction patterns (**a**); SEM-image (**b**).

**Figure 2 materials-12-00981-f002:**
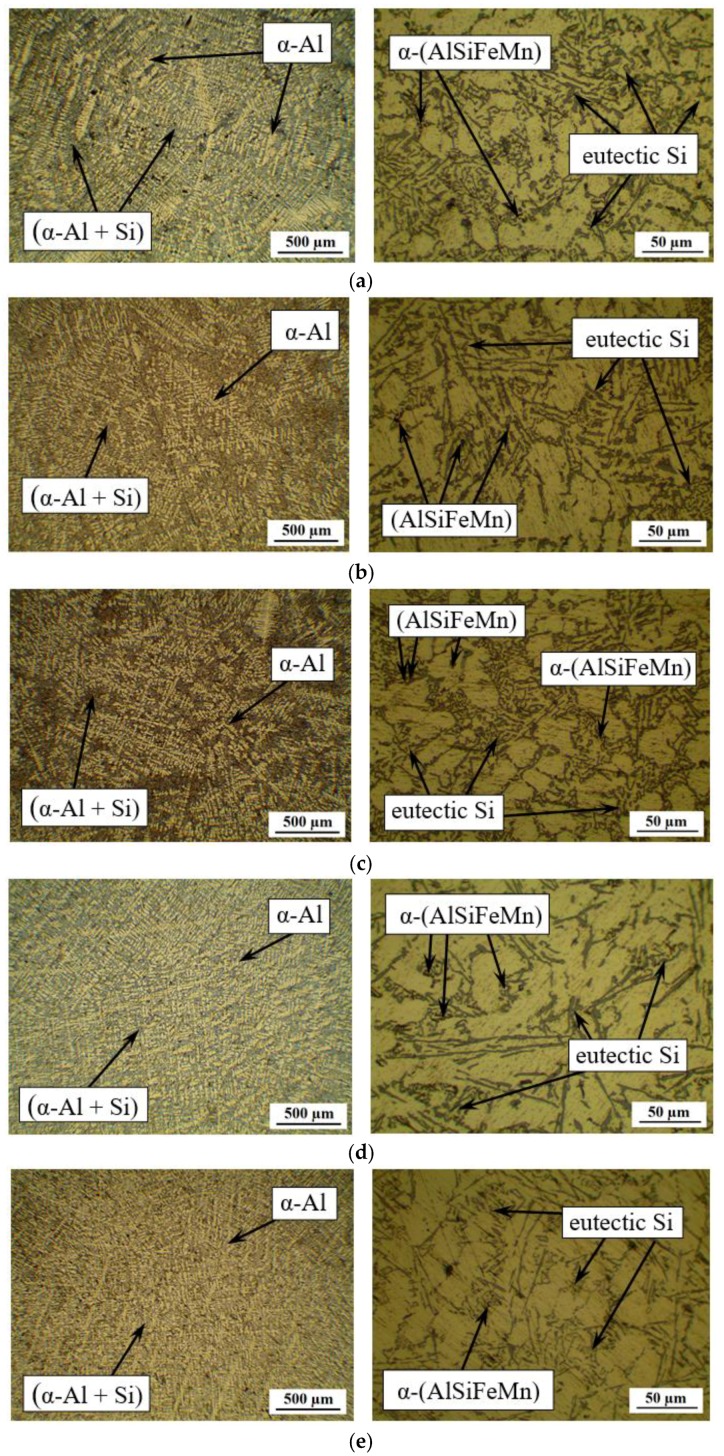
Microstructure of Al-12%Si alloy with different content of W: initial state (0 mass % of W), (**a**); 0.01 mass % of W, (**b**); 0.05 mass % of W, (**c**); 0.1 mass % of W (**d**); 0.5 mass % of W (**e**).

**Figure 3 materials-12-00981-f003:**
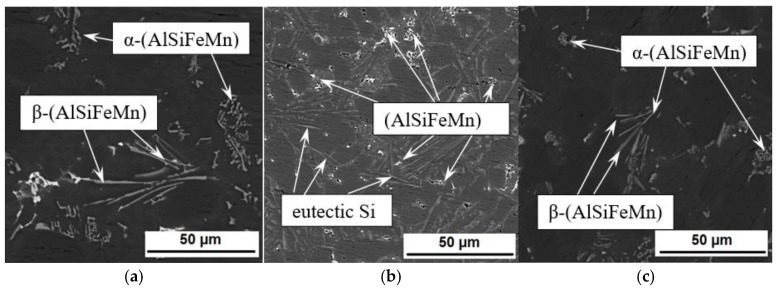
SEM-images of Al-12%Si alloys in mode of backscattered electrons: initial state (0 mass % of W), (**a**); 0.01 mass % of W (**b**); 0.1 mass % of W, (**c**).

**Figure 4 materials-12-00981-f004:**
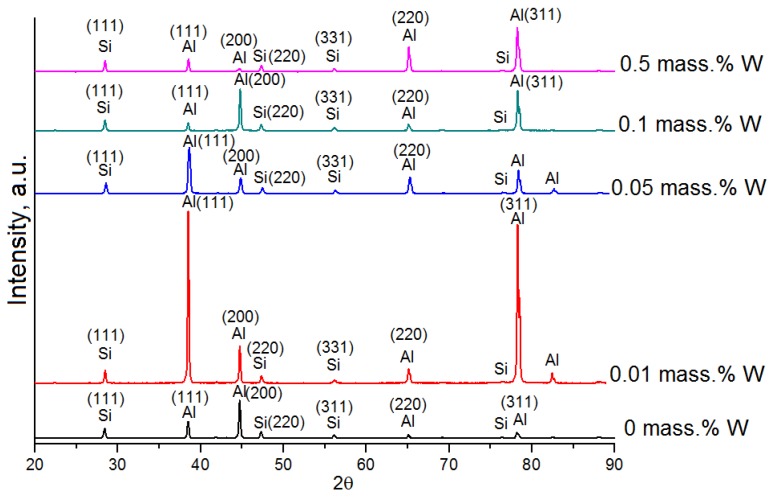
X-ray diffraction patterns of Al-12%Si alloys with different contents of W.

**Figure 5 materials-12-00981-f005:**
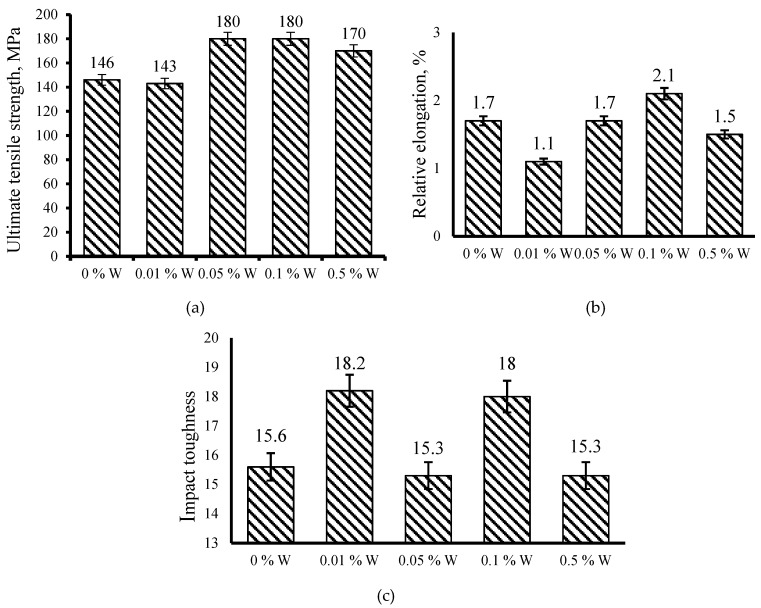
Influence of W nanopowder on properties of Al-12%Si alloy: ultimate tensile strength, MPa, (**a**); relative elongation, %, (**b**); impact toughness, (**c**).

**Figure 6 materials-12-00981-f006:**
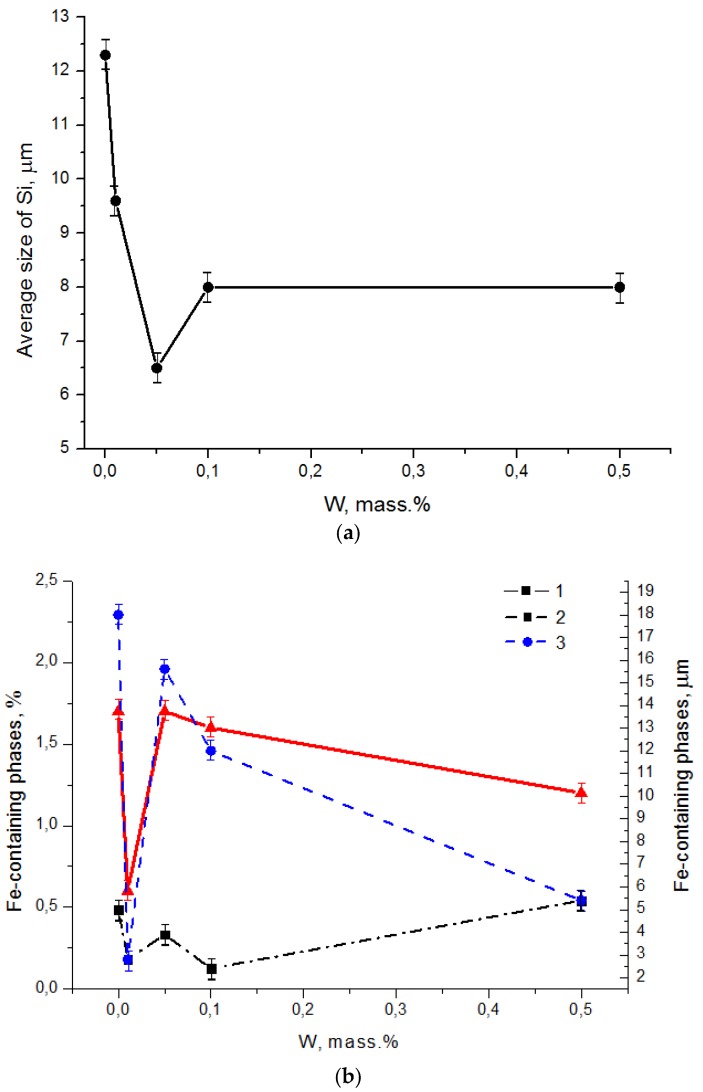
Dependence of average size of eutectic Si (**a**), volume fraction and sizes of Fe-containing phases (**b**) on W content.

**Figure 7 materials-12-00981-f007:**
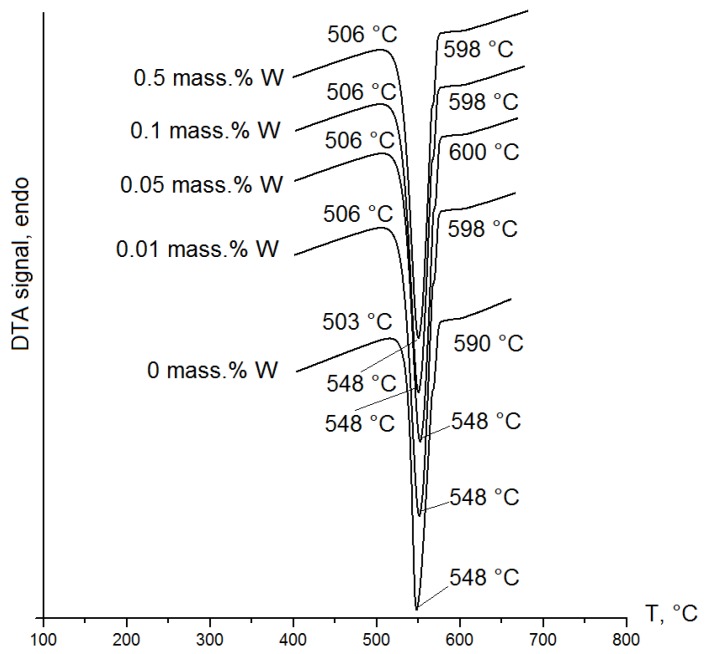
DSC-curve of Al-12%Si alloys with 0.01–0.5 mass % of W.

**Table 1 materials-12-00981-t001:** The chemical composition of the Al-12%Si alloy used in this work (mass %).

Si	Fe	Cu	Mn	Zn	Ti	Mg	Ca	Pb	Ni
11.2	0.44	0.29	0.28	0.08	0.04	0.05	0.04	0.01	0.03

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
