# Peer review of "Influence of W Addition on Microstructure and Mechanical Properties of Al-12%Si Alloys"

_materials, 2019, doi:10.3390/ma12060981_

Round 1
Reviewer 1 Report
The authors investigated the effect of W nanopowder additions on the microstructure and mechanical properties of Al-12%Si alloy. The authors give a detailed description of the alloys preparation with W (four mass%). The crystallisation process wad studied using differential scanning calorimetry. The microstructure was analysed in detail using X-ray diffraction and SEM. The ultimate tensile strength and the elongation to failure were estimated in tension at room temperature. The paper contains new and important results.
My comments:
1. The authors should give more information on tensile experiments as specimen dimensions, strain rate
2. The authors could discuss possible strengthening mechanisms.
3. The authors should use the reference style used by the journal.
Author Response
Thanks for the review and comments.
Responses to reviewer comments:
The following information on the samples has been added: Strength characteristics during tension were determined using the scheme of uniaxial tension on cylinder samples with fillets of type III with the 6 mm diameter of the working section; the diameter of the fillets (gripping parts) was 10 mm. The strain rate was 0.2 mm/sec. The changes in the text are in yellow.
This manuscript is a part of the project “Physics of formation of high-strength structural alloys based on aluminum by introduction of super-dispersed tungsten particles”. A cycle of works on modification of Al-Si alloys (with different contents of silicon) by W nanopowder is being performed. After all the studies, the preparation of a manuscript, concerning the discussion of the strengthening mechanisms, is planned.
Agree with the comment; references have been corrected.
Reviewer 2 Report
The aim of the paper, i.e. the influence of 0.01–0.5 wt. % of W nanopowder additives on microstructure, phase composition and mechanical properties of Al-12%Si alloy, is quite interesting, even if not so appealing, due to the large number of papers already published on Al-Si alloys.
The abstract summarize the work. The purpose of the study is clearly outlined and the findings of prior work are well discussed. There are no errors in logic or experimental procedure. The authors accurately explain how the data were collected. There is sufficient information that the experiment can be reproduced. All topics are well presented and discussed. The summary and conclusions are sound and justified. All presented figures are good quality and they prove their point. The paper is written in good English. The manuscript is easily readable concerning language, style and presentation. The references are appropriate and up to date.
Author Response
Thanks for the review.
Reviewer 3 Report
The paper describes the changes in the microstructure and the mechanical properties of an AlSi12-alloy due to the addition of different amounts of W Nano powder. The addition of W-powder into the melt leads to a refinement of the microstructure and an increase of the mechanical properties of the alloy.
The paper is written well and understandable but requires some minor improvements.
There are some open questions:
Line 62, 63: After melting the Al-12%Si alloy, the crucible was withdrawn from the furnace, the surface of the melt was cleaned from the oxide film and then W was introduced.
There is no information how the W powder was introduced into the melt. Was the melt stirred after introducing the W-powder to avoid clustering?
Line 98-99: According to the histogram of Si plates distribution by sizes, the average size is 12.3 μm (Figure 2a).
There is no histogram given in the paper. Some information about the measurement of the size would be good (how it was measured, how many plates have been measured,…)
Line 16: “powder W” replace by W-powder
Line 18: “of the Al-12%Si alloy” replace by of an Al-12%Si alloy
Line 19-20: “a 1.5 time decrease of the plates of eutectic Si” insert “in the size” after decrease
Line 39: “[10,11]has been” Insert a space after [10,11]
Line 60: “in the muffle” replace by “in a muffle“
Line 61: „800 ºС“ replace by „800°С“
Line 64: „melt was, mass. %: 0.01; 0.05; 0.1; 0.5.“ replace by „melt was 0.01; 0.05; 0.1; 0.5 mass. %.“
Line 65: „800 ºС“ replace by „800°С“
Line 68: “80–90 ºC/s” replace by “80–90°C/s”
Line 76: were studied by the method of differential scanning
Line 78: “using the Brinell hardness” “using a Brinell hardness”
Line 79: “with the load of 250 kg” “with a load of 250 kg”
Line 82: “calculated” replace by “measured” or “determined”
Line 83: with the computer control
Line 82: “with the 6 mm diameter of the working section.” replace by “with a diameter of 6 mm in the working section.”
Line 82: a maximum reserve of the energy
Line 88-89: “served as objects of researches.” Replace by “were used for the tests.”
Line 94: by the α-Al dendrites with the
Line 95: “and the coarse (α-Al + Si) eutectic with the coarse” Replace by “and a coarse (α-Al + Si) eutectic with a coarse”
Line 107-108: “along the boundaries of which there are fine Si particles with the size of ~1 μm.” Replace by “along the boundaries fine Si particles with the size of about 1 μm are found.”
Line 111: “leads to the uniform” replace by “leads to a uniform”
Line 124: “phase 4–45 μm”
Line 126-127: “form of a ~2.8 μm size” replace by “with a size of ~2.8 μm”
Line 130: “but the phases size is reduced more than 2 times” replace by “but the size is reduced by more than 2 times”
Line 132: presented in the subsequent
Figure 3: “in mode of reflected electrons” replace by in mode of backscattered electrons
Line 137: only the changes
Line 144: “wafers size as much as 1.5–2 times” replace by “wafers size by 1.5–2 times”
Line 145-148: The introduction of W in the amount of 0.01–0.05 mass. % results in merely the reduction of the plates of eutectic silicon; at that, as well as in the initial alloy, the clusters of (α-Al + Si) eutectic and their nonuniform distribution over the casting section are observed.
Sentence should be revised
Figure 5 (c) the unit of the Impact toughness is missing
Line 161 ff: “Figure 5 shows that the introduction of the addition in the amount of 0.1 mass. % is optimal relatively mechanical properties as it leads to the increase of the ultimate tensile strength, the relative elongation and the impact toughness by 16-20 %.” Replace by “Figure 5 shows that the introduction of the addition in the amount of 0.1 mass. % is optimal as it leads to an increase of the ultimate tensile strength, the relative elongation and the impact toughness by 16-20 %.”
Line 165: “remain without changes.” Replace by “remain unchanged.” or “remain constant.”
Line 166: “introduction of the tungsten”
Line 169ff: “The enhancement of mechanical properties of the Al-12%Si alloy when introducing W is related to a reduction of eutectic Si sizes. Figure 6a shows that with the increase of the W content, the average size of eutectic Si reduces.” Replace by “The enhancement of mechanical properties of the Al-12%Si alloy when introducing W is related to a reduction of the average size of eutectic Si (Figure 6a).”
Line 172: “and the more than 2 times reduction of sizes of Fe-containing phases” replace by “and the reduction of sizes of Fe-containing phases by more than 2 times”
Line 175: “Fe-containing phases reduces more than 2 times” replace by “Fe-containing phases is reduced to less than the half”
Line 176: “and the size reduces 2 times” replace by “and their size is reduced by 2 times”
Line 185: what means “the RE-containing intermetallic compound”
Line 188: “into the Al–20%Si alloy” replace by “into an Al–20%Si alloy”
Line 190: “of the W powder“
Line 195: “amount of the tungsten nanopowder”
Line 202: “Conducting the DSC-analysis”
Line 206, 207: “peak corresponding to eutectic Si observed” replace by “peak corresponding to the eutectic crystallization of Si is observed”
Figure 7: It would be good to color the curves
Line 212: “introduction of the W nanopowder”
Line 220ff: “In the paper [8], the authors also demonstrated that with the increase of the Sm content in the ADC12 alloy, the temperature of primary crystallization changes by 2 °С per each introduction of the addition, and the wafers of eutectic Si are refined without alteration of its morphology.” Sentence is unclear, please revise
Line 224ff: “Thus, the studies have demonstrated that the W nanopowder is efficient for refinement of structural constituents of Al-Si alloys, as, being a refractory element, it forms points of the crystallization onset in the melt shifting the temperature of the eutectic precipitation onset to the region of higher temperatures and accelerating the cooling rate of castings and refining the structural constituents of the alloy.” Sentence is unclear, please revise
Line 231: “properties of the Al-12%Si alloy” replace by “properties of an Al-12%Si alloy”
Line 234: “the reduction of Si sizes,” replace by “the reduction of the size of Si plates,”
Author Response
Thanks for the review and comments.
Responses to reviewer comments:
Line 62, 63: After melting the Al-12%Si alloy, the crucible was withdrawn from the furnace, the surface of the melt was cleaned from the oxide film and then W was introduced.
There is no information how the W powder was introduced into the melt. Was the melt stirred after introducing the W-powder to avoid clustering?
The W powder was wrapped in aluminum foil and introduced into the melt. Later, the melt was stirred during 5 min by the ceramic stirrer. The changes in the text are in blue.
Line 98-99: According to the histogram of Si plates distribution by sizes, the average size is 12.3 μm (Figure 2a).
There is no histogram given in the paper. Some information about the measurement of the size would be good (how it was measured, how many plates have been measured,…)
The sentence “According to the histogram of Si plates distribution by sizes …” has been removed from the manuscript, since the histograms were not supposed to be in the manuscript. Inclusion of the histograms would overload the manuscript with images. The silicon plates were calculated by means of the program “Atlas” provided with the software for SEM.
Line 16: “powder W” replaced by W-powder
replaced
Line 18: “of the Al-12%Si alloy” replaced by of an Al-12%Si alloy
replaced
Line 19-20: “a 1.5 time decrease of the plates of eutectic Si” complemented with “in the size” after decrease
Inserted
Line 39: “[11,12] has been” inserted a space after [11,12]
inserted
Line 60: “in the muffle” replaced by “in a muffle“
replaced
Line 61: „800 ºС“ replaced by „800°С“
replaced
Line 64: „melt was, mass. %: 0.01; 0.05; 0.1; 0.5.“ replaced by „melt was 0.01; 0.05; 0.1; 0.5 mass. %.“
replaced
Line 65: „800 ºС“ replaced by „800°С“
replaced
Line 68: “80–90 ºC/s” replaced by “80–90°C/s”
replaced
Line 76: were studied by the method of differential scanning
revised
Line 78: “using the Brinell hardness” replaced by “using a Brinell hardness”
replaced
Line 79: “with the load of 250 kg” replaced by “with a load of 250 kg”
replaced
Line 82: “calculated” replaced by “measured” or “determined”
replaced
Line 83: with the computer control
revised
Line 82: “with the 6 mm diameter of the working section.” replaced by “with a diameter of 6 mm in the working section.”
replaced
Line 82: a maximum reserve of the energy
replaced
Line 88-89: “served as objects of researches.” replaced by “were used for the tests.”
replaced
Line 94: by the α-Al dendrites with the
replaced
Line 95: “and the coarse (α-Al + Si) eutectic with the coarse” replaced by “and a coarse (α-Al + Si) eutectic with a coarse”
replaced
Line 107-108: “along the boundaries of which there are fine Si particles with the size of ~1 μm.” replaced by “along the boundaries fine Si particles with the size of about 1 μm are found.”
replaced
Line 111: “leads to the uniform” replaced by “leads to a uniform”
replaced
Line 124: “phase 4–45 μm”
replaced
Line 126-127: “form of a ~2.8 μm size” replaced by “with a size of ~2.8 μm”
replaced
Line 130: “but the phases size is reduced more than 2 times” replaced by “but the size is reduced by more than 2 times”
replaced
Line 132: presented in the subsequent
replaced
Figure 3: “in mode of reflected electrons” replaced by in mode of backscattered electrons
replaced
Line 137: only the changes
replaced
Line 144: “wafers size as much as 1.5–2 times” replaced by “wafers size by 1.5–2 times”
Line 145-148: The introduction of W in the amount of 0.01–0.05 mass. % results in merely the reduction of the plates of eutectic silicon; at that, as well as in the initial alloy, the clusters of (α-Al + Si) eutectic and their nonuniform distribution over the casting section are observed.
Sentence should be revised
revised
Figure 5 (c) the unit of the Impact toughness is missing
Line 161 ff: “Figure 5 shows that the introduction of the addition in the amount of 0.1 mass. % is optimal relatively mechanical properties as it leads to the increase of the ultimate tensile strength, the relative elongation and the impact toughness by 16-20 %.” Replaced by “Figure 5 shows that the introduction of the addition in the amount of 0.1 mass. % is optimal as it leads to an increase of the ultimate tensile strength, the relative elongation and the impact toughness by 16-20 %.”
replaced
Line 165: “remain without changes.” Replaced by “remain unchanged.” or “remain constant.”
replaced
Line 166: “introduction of the tungsten”
replaced
Line 169ff: “The enhancement of mechanical properties of the Al-12%Si alloy when introducing W is related to a reduction of eutectic Si sizes. Figure 6a shows that with the increase of the W content, the average size of eutectic Si reduces.” Replace by “The enhancement of mechanical properties of the Al-12%Si alloy when introducing W is related to a reduction of the average size of eutectic Si (Figure 6a).”
replaced
Line 172: “and the more than 2 times reduction of sizes of Fe-containing phases” replaced by “and the reduction of sizes of Fe-containing phases by more than 2 times”
replaced
Line 175: “Fe-containing phases reduces more than 2 times” replaced by “Fe-containing phases is reduced to less than the half”
replaced
Line 176: “and the size reduces 2 times” replaced by “and their size is reduced by 2 times”
replaced
Line 185: what means “the RE-containing intermetallic compound”
RE - Rare Earth
Line 188: “into the Al–20%Si alloy” replaced by “into an Al–20%Si alloy”
replaced
Line 190: “of the W powder“
replaced
Line 195: “amount of the tungsten nanopowder”
replaced
Line 202: “Conducting the DSC-analysis”
replaced
Line 206, 207: “peak corresponding to eutectic Si observed” replaced by “peak corresponding to the eutectic crystallization of Si is observed”
replaced
Figure 7: It would be good to color the curves
As is
Line 212: “introduction of the W nanopowder”
replaced
Line 220ff: “In the paper [9], the authors also demonstrated that with the increase of the Sm content in the ADC12 alloy, the temperature of primary crystallization changes by 2 °С per each introduction of the addition, and the wafers of eutectic Si are refined without alteration of its morphology.” Sentence is unclear, please revise
revised
Line 224ff: “Thus, the studies have demonstrated that the W nanopowder is efficient for refinement of structural constituents of Al-Si alloys, as, being a refractory element, it forms points of the crystallization onset in the melt shifting the temperature of the eutectic precipitation onset to the region of higher temperatures and accelerating the cooling rate of castings and refining the structural constituents of the alloy.” Sentence is unclear, please revise
revised
Line 231: “properties of the Al-12%Si alloy” replaced by “properties of an Al-12%Si alloy”
replaced
Line 234: “the reduction of Si sizes,” replaced by “the reduction of the size of Si plates,”
replaced
Reviewer 4 Report
Comments and Suggestions for improvement:
The manuscript reports on the influence of W addition on microstructure and mechanical properties of silumins.
The paper contains interesting new experimental results and the conclusions seem to be fully supported by the reported results. However, despite the clear and concise presentation, this article needs additional information to improve understanding.
Thus I recommend publication in Materials after the authors have considered the following remarks:
Could you add errors bars in Figures 5 and 6, and specify in the legends how many samples are tested?
Could you add information on how the phase identification method was performed? is it only by XRD?
For each phase composition, are EDX measurements made to locate their dispersion in the samples? Or is it only Figures 1 and 2 that allow you to conclude on their location? It is not clear in the text, could you add some details?
Author Response
Thanks for the review and comments.
Responses to reviewer comments:
Errors added.
Using XRD, it is possible to interpret only basic phases in the alloy: a-Al and Si. Therefore, to identify Fe-containing phases, we used data obtained by energy dispersive X-ray spectroscopy (EDX) compatible with a scanning electron microscope. The result of the EDX analysis is the concentrations of elements present on the surface under study with indication of confidence limit or standard deviation for the measured value of the component content. The formula was determined by recalculating the elements concentrations using proportions. Quantitative analysis of Fe-containing phases in Al-SI alloys of different alloying systems (based on phase diagrams) was studied by professor Belov and his colleagues in detail. In the journal “Foundry”, in a number of articles, authors analysed multi-component systems based on aluminum, calculated melting diagrams, determined concentration boundaries of appearance of primary crystals of Fe-containing phases. For each formula of Fe-containing phases, the ranges of possible concentrations for each element, included into the formula, were defined [Belov N.A. Quantitative analysis of the primary crystallization of iron-containing phases in relation to aluminum alloys of different doping systems. Foundry. 2013. No. 3 pp. 37-43]. To determine the formula of Fe-containing phases in our manuscript, we recalculated EDX analysis data and compared these data with the data obtained by the authors [Belov N.A. Quantitative analysis of the primary crystallization of iron-containing phases in relation to aluminum alloys of different doping systems. Foundry. 2013. No. 3 pp. 37-43].
Yes, EDX analysis was carried out for each of the phase composition. For Fe-containing phases we defined morphology and the formula was determined according to p. 2. Then we determined the sizes of these phases using the program “Atlas” provided with the software for SEM. The obtained data were compared with the data of other authors; the full-fledged analysis is presented in the manuscript with references to the works by other authors.